



# Observation-based Analysis of Ozone Production Sensitivity for Two Persistent Ozone Episodes in Guangdong, China

Kaixiang Song[1], Run Liu[1,2], Yu Wang[1,2], Tao Liu[1], Liyan Wei[1], Yanxing Wu[1], Junyu Zheng[1,2], Boguang Wang[1,2], Shaw Chen Liu[1,2]

[1]Institute for Environmental and Climate Research, Jinan University, Guangzhou, 511443, China
[2]Guangdong-Hongkong-Macau Joint Laboratory of Collaborative Innovation for Environmental Quality, Guangzhou, 511443, China

*Correspondence to*: Run Liu (liurun@jnu.edu.cn), Shaw Chen Liu (shawliu@jnu.edu.cn)

**Abstract.** An observation-based method (OBM) is developed to investigate the sensitivity of ozone formation to precursors

during two persistent elevated ozone episodes observed at 77 stations in Guangdong. Average OH concentrations derived at the 77 stations between 08:00 and 13:00 local time stay within a narrow range of $2.5 \times 10^6$ cm$^{-3}$ to $5.5 \times 10^6$ cm$^{-3}$ with a weak dependence on the NOx. These values are in good agreement with OH values observed at a rural station in Pearl River Delta (PRD) and a box model constrained by the ambient conditions. They also agree well with a box model constrained by the ambient conditions observed during the two episodes. The OBM has been used to evaluate the ozone production efficiency,

$\varepsilon(NO_x$ or VOC), defined as the number of $O_3$ molecules produced per molecule of $NO_x$ (or VOC) oxidized. Average values of $\varepsilon(NO_x)$ and $\varepsilon(VOC)$ determined by the OBM are 3 and 2.1 ppb/ppb, respectively, both compared well with values in previous studies. Approximately 67% of the station-days exhibit ozone formation sensitivity to $NO_x$, approximately 20% of the station-days are in the transitional regime sensitive to both $NO_x$ and VOC, only approximately 13% of the station-days are sensitive to VOC. These results are in semi-quantitative agreement with the ozone formation sensitivity calculated by the

box model constrained by ambient conditions observed during the two episodes. However, our OBM results differ from those of most previous investigations which suggested that limiting the emission of VOC rather than $NO_x$ would be more effective in reducing ozone reduction in Guangdong.

## 1 Introduction

Increases of surface ozone ($O_3$) can have serious adverse impacts on human health and ecological systems (Wang et al., 2005;

Song et al., 2017; Lin et al., 2018). In addition, tropospheric ozone is a significant greenhouse gas (IPCC, 2013). With a high rate of urbanization and industrialization, and the increasing use of motor vehicles, Guangdong has been suffering from severe $O_3$ pollution (Zhang et al., 2011). The primary pollutant in Guangdong has switched from particle matters to $O_3$ since 2015, thanks to a stringent emission control policy that has effectively reduced other air pollutants (Department of Ecology and Environment of Guangdong Province, 2016). In fact, the number of days with $O_3$ as the primary pollutant is 68.7%, far



exceeding that of PM$_{2.5}$ (15.8%) and PM$_{10}$ (8.3%) in 2020 (Department of Ecology and Environment of Guangdong Province, 2021).

O$_3$ is a secondary pollutant produced from photochemical reactions involving nitrogen oxides (NO$_x$) and volatile organic compounds (VOCs) (Trainer et al., 2000; Zhang et al., 2014; Wang T. et al., 2017). Sensitivity of O$_3$ production is nonlinearly dependent on precursor concentrations, and is usually categorized into photochemical regimes such as NO$_x$-

limited or VOC-limited (Kleinman et al., 1994; Sillman et al., 1998). There have been a number of studies on the sensitivity of O$_3$ production to NO$_x$ and VOC by photochemical air quality models (Sillman et al., 2003; Lei et al., 2004; Tang et al., 2010), as well as observation-based methods (OBM) (Thielmann et al., 2002; Zaveri et al., 2003; Shiu et al., 2007). Several modeling approaches have been used to evaluate the O$_3$ production sensitivity, including L$_N$/Q method, where L$_N$ is the radical loss via the reactions with NO$_x$ and Q is the total primary radical production (Kleinman et al., 2001; Kleinman, 2005;

Mao et al., 2010), the relative incremental reactivity method (RIR) (Shao et al., 2009; Cheng et al., 2010; Lu et al., 2010; Xue et al., 2014; Li et al., 2017) and the Empirical Kinetics Modeling Approach method (EKMA) (Dodge, 1977). These model-based studies usually have large uncertainties in their input parameters, particularly in the emission inventories and photochemistry of VOC (Chang et al., 2020). Observation-based methods can avoid some of the uncertainties by using observations to constrain the analysis (Thielmann et al., 2002; Zaveri et al., 2003; Shiu et al., 2007).

In this study, we adopt the approach proposed by Shiu et al. (2007) and develop an OBM to evaluate the O$_3$ production sensitivity during two multi-day O$_3$ pollution episodes in Guangdong. In this OBM, the concentration of OH is derived from observed NO$_x$ and CO in a new approach as described in the methodology section. The OBM is then used to evaluate the ozone production efficiency, ε(NO$_x$ or VOC), defined as the number of O$_3$ molecules produced per molecule of NO$_x$ (or VOC) oxidized. Finally, 3D-EKMA plots are generated basing on the OBM. The rest of the paper is organized as follows:

Section 2 describes the data sources and analysis methods, Section 3 presents the results and discussions, and Section 4 presents a summary and conclusions.

## 2 Data and methodology

### 2.1 Data

Hourly surface O$_3$, PM$_{2.5}$, CO and NO$_2$ concentration data at 77 out of a total 102 stations in Guangdong operated by China

National Environmental Monitoring Centre (CNEMC) during the period 2018–2019 are used in this study (available at http://www.cnemc.cn/en/). The 77 stations (Figure 1) are chosen for their completeness of data. It can be seen in Fig. 1 that polluted stations are mainly located in the PRD, while clean stations are located in the northeast of Guangdong. In this study, we choose two persistent O$_3$ pollution episodes to perform the OBM analysis, specifically October 2 to October 8, 2018 and September 24 to October 1, 2019.

Hourly meteorological data are obtained from European Centre for Medium-Range Weather Forecasts ERA5 reanalysis, including relative humidity (RH), 2 m temperature (T), 10 m zonal wind (U10), 10 m meridional wind (V10), geopotential

height, cloud cover, surface net solar radiation, boundary layer height and K-index, with a resolution of 0.25°×0.25° (available at https://www.ecmwf.int/).

## 2.2 Methods

### 2.2.1 Derivation of nitric oxide (NO) concentration

Since the observation of NO concentration ([NO]) is severely limited in CNEMC dataset, we calculate [NO] from the assumption of photochemical equilibrium between [NO$_2$] and [O$_3$] according to the following equation:

$$[NO]=\frac{J_{NO_2} \times [NO_2]}{[O_3] \times K_1} \tag{1}$$

where $k_1$ represents the reaction rate constant for the reaction of NO with O$_3$. This equation neglects the reactions of NO

reactions with HO$_2$ and RO$_2$. The uncertainty due to this neglection is around 20%, which is acceptable as discussed in Section 3.5. The value of $k_1$ is taken from Seinfeld and Pandis (1998):

$$k_1 (1/ppm/min)=3.23 \times 10^3 \exp(-1430/T) \tag{2}$$

$J_{NO_2}$ is the photolysis rate of NO$_2$. Its value depends on the solar zenith angle ($\chi$) and total shortwave radiation (TSR) (Wiegand and Bofinger, 2000):

$$J_{NO_2} = \begin{cases} TSR \times \left[(4.23 \times 10^{-4})+1.09 \times \frac{10^{-4}}{\cos \chi}\right] & 0°{\leq}\chi{\leq}47° \\ TSR \times (5.82 \times 10^{-4}) & 47°{<}\chi{\leq}64° \\ TSR \times \left[(-0.997 \times 10^{-4})+(1.2 \times 10^{-3}) \times (1-\cos \chi)\right] & 64°{<}\chi{\leq}90° \end{cases} \tag{3}$$

### 2.2.2 Derivation of VOC

In this study, we use CO as a tracer to estimate VOC. This tracer method has been widely used in previous studies (Heald et al., 2003; Hsu et al., 2010; Shao et al., 2011; Yao et al., 2012; Tang et al., 2013). Individual VOC at 08:00 are calculated by multiplying the freshly emitted CO at 08:00 with the ratio of VOC/CO in the emission inventories of Huang et al. (2021).

The freshly emitted CO is assumed to be the difference in CO between 08:00 and 13:00 (Fig. 2). The CO at 13:00 is considered to be the leftover CO for the following day, and is in turn used to evaluate the leftover VOC. Oxidized VOC (OVOC) are estimated from the observed ratios of CH$_2$O, CH$_3$CHO and ketone to CO (Wang et al., 2016). Other OVOCs are not included.

### 2.2.3 Derivation of OH concentrations

The ratio ethylbenzene/m,p-xylene has been suggested to be a good measure of the photochemical processing by OH (Calvert, 1976; Singh, 1977; Shiu et al., 2007). Following a Lagrangian trajectory, the ratio can be shown as

$$E/X =(E_0/X_0)\exp\left(- \int_0^t (k_e - k_x) [OH] \, dt\right) \tag{4}$$



Where E and X represent concentrations of ethylbenzene and m,p-xylene at time t, respectively. $E_0$ and $X_0$ are their corresponding initial concentrations, $k_x$ and $k_e$ are their reaction rate constants with OH, $k_x$ and $k_e$ equal to $2.17\times10^{-11}$ and

$7.0\times10^{-12}$ cm$^3$ s$^{-1}$, respectively (Atkinson, 1990). With known value of $E_0/X_0$, [OH×t] can be evaluated from observed E/X at time t. This provides an OBM derived density of OH.

In the real atmosphere, the Lagrangian condition rarely exists due to turbulence mixing as well as atmospheric advection. Nevertheless, Equation (4) tends to hold because atmospheric transport affects the two species similarly. This is a key advantage of the OBM. In this study, due to limited measurements of VOC, we use CO and NO$_x$ to replace E and X,

respectively.

### 2.2.4 Calculation of oxidized VOC and NOx

In this study, we consider reaction of NO$_2$ with OH as the only removal process for NO$_x$ and assume the removal of NO$_x$ is pseudo-first order as shown below. In this case, following the Lagrangian trajectory, we have:

$$[NO_x]=[NO_x]_0\exp\left(-k\int_0^t[OH]\,dt\right) \qquad (5)$$

where k is the reaction rate constant of NO$_x$ with OH. The reaction rate constant for NO$_2$ and OH is $1.04\times10^{-11}$ cm$^3$ s$^{-1}$ at 25 °C and 1 atm pressure according to Sander et al. (2003). Since NO$_2$ is part of NO$_x$, the value of k should be scaled down by the ratio NO$_2$/NO$_x$. The average of NO$_2$/NO$_x$ ratio is about 0.6, thus k for NO$_x$ is prescribed at $6.0\times10^{-12}$ cm$^3$ s$^{-1}$.

Similarly, we have:

$$[VOC_s]=[VOC_s]_0\exp\left(-k_{voc_s}\int_0^t[OH]\,dt\right) \qquad (6)$$

$$[CO]=[CO]_0\exp\left(-k_{co}\int_0^t[OH]\,dt\right) \qquad (7)$$

where $k_{voc}$ and $k_{co}$ are the reaction rate constant of VOCs and CO with OH, respectively. $K_{VOC}$ of individual VOC are listed in Table S1 and $k_{CO}$ is prescribed at $1.4\times10^{-13}$ cm$^3$ s$^{-1}$ (Atkinson et al., 2006).

Since the Lagrangian condition is sometimes not observed, it is necessary to select the time periods during which the quasi-Lagrangian condition as shown in Fig. 2 is valid approximately. The selection criterion is that the ratio of CO concentrations

between 08:00 and 13:00 lies within one half of 1-standard deviation of the ratio of CO shown in Fig. 2 which is assumed to be in the Lagrangian condition. This criterion usually filters out about 60% of data, i.e., about 40% of the days satisfy approximately the Lagrangian condition.

### 2.2.5 Dilution effect

Diurnal variations of pollutants averaged over all stations and the two episodes are shown in Fig. 2. Previous studies have

shown that part of the early morning rise in O$_3$ is due to O$_3$ entrained from the residual layer above the boundary layer during the development of the boundary layer in the morning (Shiu et al., 2007; Zhao et al., 2019). We adopt the approach proposed by Shiu et al. (2007) to account for the dilution effects. Specifically, the reduction of CO concentrations from 08:00 to 13:00



(approximately 20%) is assumed to be the dilution effect, and used for all other species. The uncertainty due to this assumption is discussed in section 3.5.

### 2.2.6 Emissions of NOx, CO and VOC between 08:00 and 13:00

Equations (5), (6) and (7) do not account for the emissions of $NO_x$, CO or VOC during the period of 08:00–13:00. Inclusion of these emissions would increase the value of OH derived from Equation (5) as well as the dilution effect. We estimate the emission of $NO_x$ by taking advantage of the fact that $NO_x$ reaching a quasi-steady state around 13:00–16:00 as evident in Fig. 2. We believe that the quasi-steady state is maintained by the balance between the oxidation of $NO_x$ and its emission. Using the average OH of $5\times10^6$ $cm^{-3}$ at noontime derived from Equation (5) (Fig. 5) and mean $NO_x$ in 13:00–16:00 (Fig. 2), a value of approximately 1.8 ppb per hour can be obtained. This value is assumed to be the hourly NOx emission rate between 08:00 and 13:00. The emissions of CO and VOC are calculated using their ratios to $NO_x$ in the emission inventories of Huang et al. (2021).

### 2.2.7 Box model

A photochemical box model with carbon bond mechanism (PBM-CB05) (Yarwood et al., 2005; Coates and Butler, 2015; Wang Y. et al., 2017) is used to simulate the $O_3$ production rate and OH radical. Unlike emission-based models, the PBM-CB05 used in this study is based on observed concentrations of air pollutants and meteorological parameters (Wang Y. et al., 2017). In the CB05 module, VOCs are grouped according to carbon bond type and the reactions of individual VOCs are condensed using the lumped structure technique (Yarwood et al., 2005; Coates and Butler, 2015). In this study, the pollution indicators ($O_3$, NO, $NO_2$, CO and VOC) and meteorological parameters (temperature, relative humidity, pressure) observed during the two episodes are utilized as input parameters for the model. There are 37 VOC species considered in our case. The model simulation starts from 07:00 and end at 18:00 with hourly input data based on observed concentrations of air pollutants and meteorological parameters during the two episodes.

## 3 Results and discussion

### 3.1 Air quality and meteorological conditions

Figure 3 shows the time series of hourly concentrations of air pollutants. The time period covers the two ozone episodes and extends to two days before and two days after. Mean maximum daily 8-h average (MDA8) $O_3$ concentrations in the episode1 was 88.7 ppb, and the episode2 was 99.6 ppb. The average daily concentration of CO of the two episodes was 0.74 and 0.85 ppm, respectively. The corresponding time series of key meteorological parameters are shown in Fig. 4. As $O_3$ is formed through photochemical reactions involving precursors $NO_x$ and VOC, strong solar radiation, high temperature and low wind speed have been identified to be common conditions conducive to the formation of ozone (Liu et al., 2017; Wang L. et al., 2018). During both $O_3$ episodes, the weather in Guangdong was dominated by high pressure systems with warm and





cloudless conditions, and northeasterly winds. In particular, the average maximum temperature for the episode1 was 28℃ and the episode2 was 30℃.

The general patterns of $O_3$ concentrations of the two episodes were similar. Relatively high $O_3$ concentrations with north or northeasterly winds appeared at least two days before the episode in both episodes. Afterward, the high $O_3$ kept increasing or stayed at a high level until the prevailing northeasterly wind shifted away and the surface pressure dropped. Starting on September 22, 2018, a precipitation event occurred which obviously ended the first episode. The heavier cloud cover greatly reduced the intensity of solar radiation and $O_3$ photochemical formation reactions. The disappearance of high $O_3$ in the

second episode is believed to be related to a shift to southerly winds that brought in warm and moist air.

## 3.2 OH concentrations derived from OBM

Figure 5 shows the hourly OH concentrations between 08:00 and 13:00 derived from Equation (4) based on the concentrations of $NO_x$ and CO observed at the 77 stations. Average OH concentrations derived at the 77 stations between 08:00 and 13:00 local time stay within a narrow range of $2.5\times10^6$ cm$^{-3}$ to $5.5\times10^6$ cm$^{-3}$ with a weak dependence on the NOx.

The mean OH concentrations and their 1-standard deviations derived by the OBM (black dots and black vertical bars, respectively) are approximately 30% higher than the mean OH concentrations and 1-standard deviations observed at a rural station in PRD in October–November 2014 (blue line and blue shade, respectively) (Tan et al., 2019). Nevertheless, there is a nearly complete overlap of the 1-standard deviations of the two data sets (blue shade and black vertical bars), which indicates a good agreement between our OBM OH values and those observed by Tan et al. (2019). In another comparison

with a previous investigation, our OH concentrations are approximately 40% lower than the OH calculated by a box model constrained by observed air pollutants during an experiment at a remote island site in the PRD from August to November 2013 (red line and red shade) (Wang Y. et al., 2018). There is also a nearly complete overlap of the 1-standard deviations of the two data sets (red shade and black vertical bars). Figure 5 also includes the noontime OH concentrations calculated by the box model described above. The box model is constrained by the ambient conditions observed during the two episodes.

The average modelled OH concentration is approximately $3.2\times10^6$ cm$^{-3}$ with a 1-standard deviation of $0.6\times10^6$ cm$^{-3}$ (red cross and red vertical bar). This value of OH is approximately 40% less than the OH values of $(5.5\pm4.3)\times10^6$ cm$^{-3}$ derived by the OBM at noontime. Again, there is a good overlap of the 1-standard deviations of the two data sets. The agreement among the OH concentrations derived by the OBM, the box model and field observations gives credence to our observation-based analysis, at least in terms of the derived OH concentration which plays a critical role in the $O_3$ formation.

Nevertheless, we acknowledge that the OH concentrations derived here are approximately a factor of 3 to 5 lower than the OH concentrations observed at Backgarden (a suburban site about 70 km downwind of Guangzhou) during an intensive campaign in 2006, in which the OH reached daily peak values of $(15–26)\times10^6$ cm$^{-3}$ (Lu et al., 2012). This discrepancy remains unresolved.



### 3.3 Ozone production efficiency

Ozone production efficiency ($\varepsilon$) is defined as the number of $O_3$ molecules produced per molecule of $NO_x$ (or VOC) oxidized photochemically (Liu et al., 1987; Trainer et al., 2000). $\varepsilon$ can be calculated by the following equations:

$$\varepsilon[NOx] = \Delta[O_3]/\Delta[NO_x]$$
$$\varepsilon[VOC] = \Delta[O_3]/\Delta[VOC]$$

where $\Delta[O_3]$ represents the amount of ozone generated from 08:00 to 13:00 which is equal to the observed difference in $O_3$
between 08:00 and 13:00, after adjustment to the dilution factor. $\Delta[NO_x]$ ($\Delta[VOC]$) represents the consumption/oxidation of $NO_x$ (VOC) between 08:00 and 13:00.

Figures 6a shows the relationship of $\varepsilon$ as a function of the average $NO_x$ concentration between 08:00 and 13:00. As expected $\varepsilon$ is greater at lower $NO_x$, i.e., the $O_3$ production efficiency is greater in rural and suburban environments than urban conditions, in agreement with previous findings (Liu et al., 1987; Kleinman et al., 2002). The value of $\varepsilon$ converges to a
narrow range of about 1.0±0.5 when $NO_x$ is greater than 70 ppb. This range of $\varepsilon(NO_x)$ is consistent with previous investigations in urban environments (Sillman et al., 1998; Daum et al., 2000) as well as in rural environments (Chin et al., 1994; Trainer et al., 1995). Figure 6b is the same as Fig. 6a except that the x-axis is changed to $\Delta[NO_x]$ or the oxidized $NO_x$. Fig. 6b shows a relatively smoother distribution compared to Fig. 6a, most likely because that the oxidized $NO_x$, rather than $NO_x$ itself, is more closely related photochemically to $\Delta[O_3]$. As $\Delta[NO_x]$ increases beyond 30 ppb, $\varepsilon[NO_x]$ levels off linearly
to a nearly constant value around 1.0 when $\Delta[NO_x]$ approaches 80 ppb (Fig. 6b). $\varepsilon[VOC]$ is also greater at lower $\Delta[VOC]$, and has an asymptotic value of about 1.0±0.5 when $\Delta[VOC]$ gets greater than 50 ppb (Fig. 6c).

Figures 6b and 6c have some useful implications for the ozone control strategy. For instance, $\varepsilon[NO_x]$ = 1.7 when $\Delta[NO_x]$ = 50 ppb can be interpreted as - in a highly polluted ambient environment in Guangdong where $\Delta[NO_x]$ equals 50 ppb, approximately 1.7 ppb of ozone is produced for each ppb of $NO_x$ oxidized. The overall average value of $\varepsilon[NO_x]$ is about 3
(Fig. 6b), which implies in average 3 ozone molecules are produced for each $NO_x$ molecule oxidized. The overall average value of $\varepsilon[VOC]$ is approximately 2.1 (Fig. 6c), which implies 2.1 ozone molecules are produced for each VOC molecule oxidized, about 50% less efficient than that of $NO_x$.

Photochemical oxidation of a VOC molecule under common ambient urban conditions produces approximately two or more peroxyl radicals–one $HO_2$ and more than one $RO_2$ (Seinfeld and Pandis, 1998; Jacob, 1999). Because there is abundant $NO_x$
in the ambient atmosphere in Guangdong, nearly all peroxyl radicals are expected to react with NO to produce $NO_2$ and then $O_3$. Jacob (1999) suggested an ozone formation rate of $2\Delta[VOC]$ in the urban atmosphere. This is in excellent agreement with the overall value of $2.1\Delta[VOC]$ found here by the OBM. This agreement as well as the consistency with previous investigations on the $\varepsilon[NO_x]$, provides credence again to the observation-based analysis of this study.



### 3.4 Ozone sensitivity to precursors

The sensitivity of ozone formation ($\Delta O_3$) to ozone precursor $NO_x$ is examined in Fig. 7a in which $\Delta O_3$ (right hand side in red) and the oxidized VOC (left hand side in black) are plotted as a function of the oxidized $NO_x$. Similarly in Fig. 7b $\Delta O_3$ (right hand side in red) and the oxidized $NO_x$ (left hand side in black) are plotted as a function of the oxidized VOC. It can be seen in Fig. 7a that $\Delta O_3$ increases with the value of oxidized $NO_x$. The increase first has a very sharp slope of about 2 ppb/ppb when oxidized $NO_x$ is below 30 ppb, indicating a strong sensitivity of ozone formation to oxidized $NO_x$. The slope flattens

out quickly to around 0.2 ppb/ppb when oxidized $NO_x$ gets greater than 30 ppb, suggesting other factors such as VOC and the VOC/$NO_x$ ratio may become more important in controlling the ozone formation rate. Fig. 7b shows that $\Delta O_3$ increases with the value of oxidized VOC with a slope of about 0.4 ppb/ppb. However, this slope is much smaller than that of $NO_x$, especially in the low oxidized NOx regime (<30 ppb). In a brief summary for Figs. 7a and 7b, the ozone formation is most sensitive to the oxidized $NO_x$ in relative clean regimes of oxidized $NO_x$<30 ppb. In more polluted regimes, other factors such

as the initial VOC and/or the VOC/$NO_x$ ratio appear to have a significant impact on the ozone formation. Additional evidence in support of these points is elaborated below.

Figure 8 presents a three-dimensional EKMA-like depiction of ozone formation rates ($\Delta O_3$, black dots, 471 points) plotted as a function of the oxidized $NO_x$ (x-axis) and oxidized (VOCs+CO) (y-axis). The colored plane is a linear regression to the ozone formation rates (black dots), and the green and red bars denote positive and negative deviations of individual dots

from the plane, respectively. Different color shades from blue to red denote different levels of $\Delta O_3$ in ppb. The equation for [$\Delta O_3$] represents the plane as a function of the oxidized $NO_x$ ($\Delta NO_x$) and oxidized VOC ($\Delta VOC$). The coefficients in front of $\Delta NO_x$ and $\Delta VOC$ in the equation are the ozone sensitivities to $\Delta NO_x$ and $\Delta VOC$, respectively. The plane fits the black dots (ozone formation rates) reasonably well with an $R^2$ value of 0.423. The coefficient of $\Delta NO_x$ is 0.755 which is about 3 times of that of $\Delta VOC$ (0.247), indicating the ozone formation rate is about 3 times more sensitive to $\Delta NO_x$ than $\Delta VOC$

when consider all data at the 77 stations in Guangdong during the two episodes. This is consistent with the findings from Figs. 7a and 7b.

There appear some uneven congregations of red and green bars, e.g., a large number of red bars have low values of $\Delta NO_x$, while many green bars tend to have moderate values of $\Delta NO_x$ and high values of $\Delta VOC$. This suggests that there is a need to divide Fig. 8 into different congregations/regimes. Figure 9 is the same as Fig. 8 except dividing it into four quadrants of

different levels of oxidized ozone precursors: Quadrant (a) low $\Delta NO_x$ and low $\Delta VOC$ ($\Delta NO_x$<20 ppb, $\Delta VOC$<25 ppb), Quadrant (b) high $\Delta NO_x$ and high $\Delta VOC$ ($\Delta NO_x$>20 ppb, $\Delta VOC$>25 ppb), Quadrant (c) low $\Delta NO_x$ and high $\Delta VOC$ ($\Delta NO_x$<20 ppb, $\Delta VOC$>25 ppb), Quadrant (d) high $\Delta NO_x$ and low $\Delta VOC$ ($\Delta NO_x$>20 ppb, $\Delta VOC$<25 ppb).

39% of all data points (184 out of 471 points) lie in Quadrant (a), the slope of $\Delta O_3$ against $\Delta NO_x$ (coefficient of $\Delta NO_x$ in the equation) is approximately 1.54 ppb/ppb (p value < 0.01), while the slope of $\Delta O_3$ against $\Delta VOC$ (coefficient of $\Delta VOC$) has a

value of 0.28 ppb/ppb (p value = 0.021). These values of slopes imply that the ozone formation at stations in Quadrant (a), a relatively clean environment, is about five times more sensitive to $\Delta NO_x$ than $\Delta VOC$, i.e. the ozone formation is $NO_x$-limited.





This is in good agreement with the conclusion reached based on Figs. 7a, 7b and 8. Quadrant (b) contains about 20% of the data points. The coefficient of $\Delta NO_x$ is 0.3 ppb/ppb (p value < 0.01), while the coefficient of $\Delta VOC$ is 0.29 ppb/ppb (p value = 0.043), suggesting that the ozone formation is sensitive to both $\Delta VOC$ and $\Delta NO_x$. This quadrant belongs to the transitional

regime. Quadrant (c) has 28% of the data points, the coefficients of $\Delta NO_x$ and $\Delta VOC$ are 2.25 ppb/ppb (p value < 0.01) and 0.04 ppb/ppb (p value = 0.785), respectively. Here again the ozone formation is $NO_x$-limited. Quadrant (d) has 13% of the data points, the coefficients of $\Delta NO_x$ and $\Delta VOC$ are 0.18 ppb/ppb (p value = 0.126) and 0.91 ppb/ppb (p value = 0.037), respectively. These values of coefficients indicate that the ozone formation is more sensitive to $\Delta VOC$ than $\Delta NO_x$, i.e. the ozone formation is VOC-limited.

The analysis above provides an observation-based method for evaluating the ozone-precursor sensitivity. This method has the potential to provide quantitative information on the ozone control strategy for individual regions. In theory, the quadrants can be further divided into, for example, a specific region represented by individual stations, such that an ozone control strategy suitable to the region could be developed. In practice, this is limited by the data available for making the 3-dimentional plot like Fig. 9.

We have compared the OBM results to those of the box model constrained by the observed ambient environment in this study. Figure 10 shows the traditional 2D-EKMA plot calculated by the model. To facilitate the comparison, the x-axis and y-axis in Fig. 10 are changed to hourly oxidized $NO_x$ and oxidized VOC, respectively, rather than the usual early morning concentrations of $NO_x$ and VOC. The modeled results are shown in colored isopleths of ozone increments between 06:00 and 16:00 local time, while results of the OBM are shown in colored dots for ozone increment/formation between 08:00 and

13:00. The difference in the length of time has negligible effect on the ozone increment as evident in Fig. 2. The OBM values agree with the model results semi-quantitatively. For instance, the colored dots of OBM shift from blue (20 ppb) to green (60–80 ppb) consistently with the colored isopleths, but the OBM dots rarely turn yellow when modeled isopleths get greater than 90 ppb. Two red lines (left-red and right-red) are added to Fig. 10 to facilitate the assessment of the sensitivity of ozone formation. There are 127 points located to the left of left-red line, which clearly belongs to the $NO_x$-limited regime

according to the modeled ozone isopleths. There are 141 points located to the right of right-red line, which clearly belongs to the VOC-limited regime according to the modeled ozone isopleths. In between the two red lines contains 203 points, which are in the transitional regime sensitive to both $NO_x$ and VOC. These three regimes overlap and agree in ozone formation sensitivity with Quadrants (a and c), Quadrant (d) and Quadrant (b) of the OBM results, respectively. However, the numbers of points in the three regimes deviate significantly from those of four OBM Quadrants. For example, Quadrant (b) has only

97 points compared to the 203 points in the transitional regime of Fig. 10; Quadrant (d) has only 60 points compared to the 141 points in the VOC-limited regime of Fig. 10; while Quadrants (a and c) has 314 points compared to the 127 points in in $NO_x$-limited regime of Fig. 10. In terms of ozone sensitivity, the modeled results show nearly equal number of points in the $NO_x$-limited regime as the VOC-limited regime, while the OBM results show five to one in favor of the $NO_x$-limited regime. A quantitative agreement between the OBM results (dots) and the modeling results (isopleths) would require to shift the dots

in Fig. 10 leftward by approximately 0.5–1 ppbv $h^{-1}$, which would mean a reduction of OH by approximately 30–50%.



Interestingly this requirement matches well with the fact that modeled OH is approximately 40% less than the OH value derived by the OBM at noontime as shown in Fig. 5.

Comparing with previous studies, we notice that almost all previous researches suggested that limiting the emission of VOCs in Guangdong would have a positive role in reducing ozone reduction (Zhang et al., 2008; Wang T. et al., 2017; Jiang et al.,
2018), but different results may appear in different places and time. Yu et al. (2020) found that $NO_x$ reduction in Shenzhen has led to higher ozone production from 2015 to 2018 given the nearly constant VOC. However, the ozone mitigation would be benefit from further $NO_x$ reduction under the conditions of 2018. Yang et al. (2019) analyzed the relationship between ozone and precursors in PRD from 2007 to 2017 and found that the northeastern PRD was $NO_x$-limited and the southwest VOC-limited. Obviously, these findings are in general different from our results except in highly polluted environment like
Quadrant (b). Some of the difference can be explained by the fact that most of the previous studies were focused on urban regions, while many rural stations are included in our OBM analysis. Finally, we acknowledge that our results are based on the analysis of only two multi-day ozone episodes which maybe not representative of the general ambient environment in Guangdong. A comprehensive regional and temporal OBM analysis is needed to make a definitive comparison with previous findings.

In summary of Section 3.4, the sensitivity of ozone formation to its precursors is complex and highly dependent on the ambient conditions of the station-day. Our OBM shows that approximately 67% of the station-days exhibit ozone formation sensitivity to $NO_x$, approximately 20% of the station-days are in the transitional regime sensitive to both $NO_x$ and VOC; only approximately 13% of the station-days are sensitive to VOC. These findings are different from results of most previous studies, which favor ozone formation sensitivity to VOC.

**3.5 Uncertainty analysis**

Significant uncertainties and limitations exist in our OBM analysis. First and foremost is the uncertainty involved with the Lagrangian air mass assumption, which doesn't take into account of mixing, entrainment or surface deposition effects. Omitting the mixing of $NO_x$ emitted between 08:00 and 13:00 into the Lagrangian air mass can lead to an underestimate of the OH concentration, while omitting the mixing of CO emission can underestimate the dilution effect. We account for the
mixing of $NO_x$ emission by assuming that $NO_x$ reached a quasi-steady state around 13:00–16:00 (section 2.2.6), and in turn the mixing of CO and VOC emissions are calculated using their ratios to $NO_x$ in the emission inventories of Huang et al. (2021). However, no surface deposition effect is included. The selection criterion defined by 50% of 1-standard deviation ($1.0\pm0.5\sigma$) from the mean CO distribution works well in filtering out those data deviating significantly from the Lagrangian condition. However, the criterion filters out about 60% of the data, thus limiting the representativeness of the OBM analysis.
This limitation has been evaluated by relaxing the selection criterion to $1.0\pm0.8\sigma$, which filters out only about 30% of the data. No significant difference has been detected, suggesting the results of the OBM analysis are representative of the majority of the data. Another source of uncertainty is that one single dilution factor is adopted for all air pollutants, including $O_3$, CO, $PM_{2.5}$ and $NO_x$. In this context, it is reassuring to find out that the dilution factor derived independently from CO





and PM$_{2.5}$ agrees within 10% with each other. In a brief summary, we estimate the uncertainty involved with the Lagrangian
assumption to be in the range of 20–40%.

The second largest source of uncertainty is the evaluation of VOCs. Individual VOCs at 08:00 are calculated by multiplying the fresh emission of CO at 08:00 with the ratio of VOC/CO in the emission inventories (Huang et al., 2021). The fresh emitted CO is assumed to be the difference in CO between 08:00 and 13:00. The CO at 13:00 is considered to be the leftover CO for the following day, and is in turn used to evaluate the leftover VOC. Oxidized VOC or OVOC are estimated from the
observed ratios of CH$_2$O, CH$_3$CHO and ketone to CO (Wang et al., 2016). No other OVOC is included. Another source of uncertainty is attributable to the coarse resolution of CO measurements which is reported at 0.1 ppm intervals. As a result, many hourly CO data would show identical values and lose its value as a tracer. We estimate the uncertainty in the evaluation of VOCs to be in the range of 30%.

## 4 Summary and Conclusions

In this study, two persistent elevated ozone episodes in Guangdong (77 stations) that occurred in October 2–October 8, 2018 and September 24–October 1, 2019 were analyzed to investigate the sensitivity of ozone generation to precursor concentrations at the 77 stations. An OBM is developed by modifying the approach suggested by Shiu et al. (2007). Specifically, NO$_x$ and CO are used in this OBM to substitute for the two hydrocarbon species utilized in Shiu et al. (2007). Major outputs from the OBM include the OH concentrations, O$_3$ production efficiency and the sensitivity of ozone formation
to the precursors at the 77 stations during the two ozone episodes. The average OH concentrations between 08:00 and 13:00 agree well with the OH values observed at a rural station in PRD in October–November 2014 by Tan et al. (2019). The OH values derived from the OBM are also in good agreement with a box model constrained by the ambient conditions observed during the two episodes. On the other hand, the OH concentrations derived here are approximately a factor of 2 to 4 lower than the OH concentrations observed at Backgarden, a suburban site about 70 km downwind of Guangzhou (Lu et al., 2012).
The O$_3$ production efficiency against NO$_x$, $\varepsilon(NO_x) = \Delta[O_3]/\Delta[NO_x]$, is greater at lower NO$_x$ (Fig. 6a), in agreement with previous findings (Liu et al., 1987; Kleinman et al., 2002). The value of $\varepsilon$ converges to a narrow range of about 1.0±0.5 when NO$_x$ is greater than 70 ppb. This range of $\varepsilon(NO_x)$ is consistent with previous investigations in urban environments (Sillman et al., 1998; Daum et al., 2000) as well as in rural environments (Chin et al., 1994; Trainer et al., 1995). The overall average value of $\varepsilon[NO_x]$ is about three (Fig. 6b), which implies on average three ozone molecules are produced for each NO$_x$
molecule oxidized. The overall average value of $\varepsilon[VOC]$ is approximately 2.1 (Fig. 6c), which implies 2.1 ozone molecules are produced for each VOC molecule oxidized, about 50% less efficient than that of NO$_x$. Jacob (1999) suggested an ozone formation rate of 2$\Delta[VOC]$ in the urban atmosphere. This is in excellent agreement with the value of 2.1$\Delta[VOC]$ found here by the OBM. This agreement as well as the consistency with previous investigations on the $\varepsilon[NO_x]$ and OH concentrations, provide credence to the observation-based analysis (OBM) of this study.

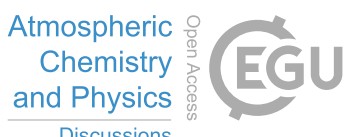

The sensitivity of ozone formation to its precursors is complex and highly dependent on the ambient conditions of the station-day. Our OBM shows that approximately 67% of the station-days exhibit ozone formation sensitivity to $NO_x$, approximately 20% of the station-days are in the transitional regime sensitive to both $NO_x$ and VOC, only approximately 13% of the station-days are sensitive to VOC. These findings are different from results of most previous studies, which favor ozone formation sensitivity to VOC. Some of the difference can be explained by the fact that most of the previous studies

were focused on urban regions, while many rural stations are included in our OBM analysis. Finally, we acknowledge that our results are based on the analysis of only two multi-day ozone episodes which maybe not representative of the general ambient environment in Guangdong. A comprehensive spatial and temporal OBM analysis is needed to make a definitive comparison with previous findings.

*Data availability.* Hourly surface $O_3$, $PM_{2.5}$, CO and $NO_2$ data were obtained from China National Environmental Centre (http://www.cnemc.cn/en/, last access: 10 November 2021). Hourly meteorological data are obtained from European Centre for Medium-Range Weather Forecasts ERA5 reanalysis (https://www.ecmwf.int/, last access: 10 November 2021). The data of this paper are available upon request to Shaw Chen Liu (shawliu@jnu.edu.cn).

*Author Contributions.* SL proposed the essential research idea. KS performed the analysis. KS, RL and SL drafted the manuscript. YW, TL, LW, YW, JZ, and BW helped analysis and offered valuable comments. All authors have read and agreed to the published version of the manuscript.

*Competing interests.* The authors declare that they have no conflict of interest.


*Acknowledgments.* The authors thank the China National Environmental Centre and European Centre for Medium-Range Weather Forecasts for providing datasets that made this work possible. We also acknowledge the support of the Institute for Environmental and Climate Research and Guangdong-Hongkong-Macau Joint Laboratory of Collaborative Innovation for Environmental Quality in Jinan University.


*Financial support.* This research was supported by the National Natural Science Foundation of China (grant number 92044302, 41805115), Guangzhou Municipal Science and Technology Project, China (grant number 202002020065), Special Fund Project for Science and Technology Innovation Strategy of Guangdong Province (grant number 2019B121205004), Guangdong Innovative and Entrepreneurial Research Team Program (grant number 2016ZT06N263),

and National Key Research and Development Program of China (grant number 2018YFC0213906).





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



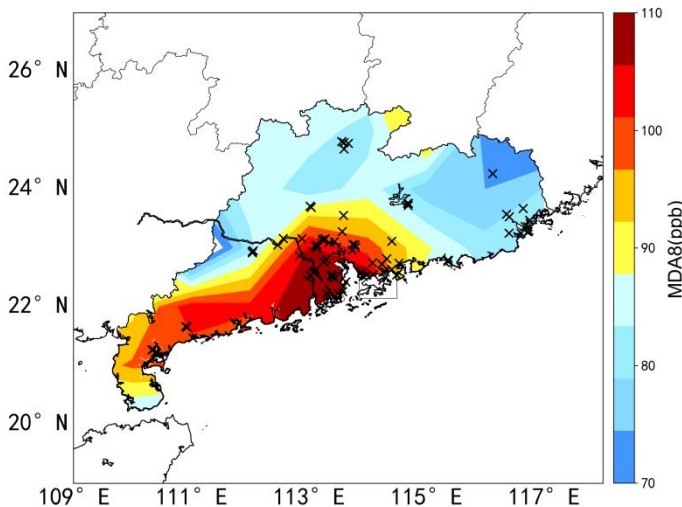

Figure 1: Spatial distribution of the average maximum daily 8-hour average ozone concentration (MDA8) in Guangdong during the study period, black crosses mark the location of the observation sites.



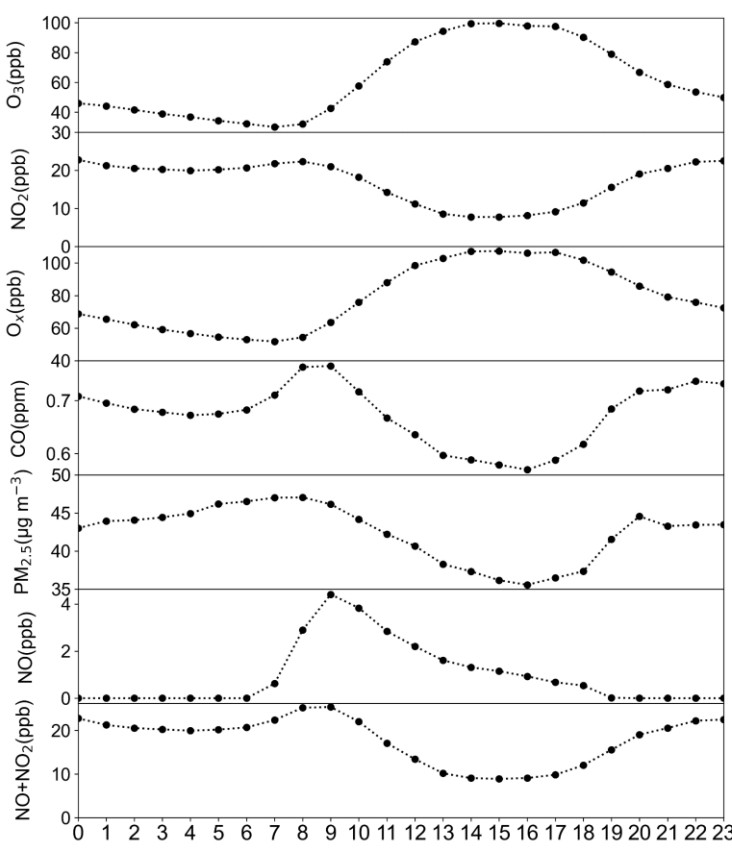

**Figure 2:** Average diurnal variations of air pollutants, $O_3$, $NO_2$, $O_x$, CO, $PM_{2.5}$, NO and $NO+NO_2$, observed in Guangdong during the two episodes.





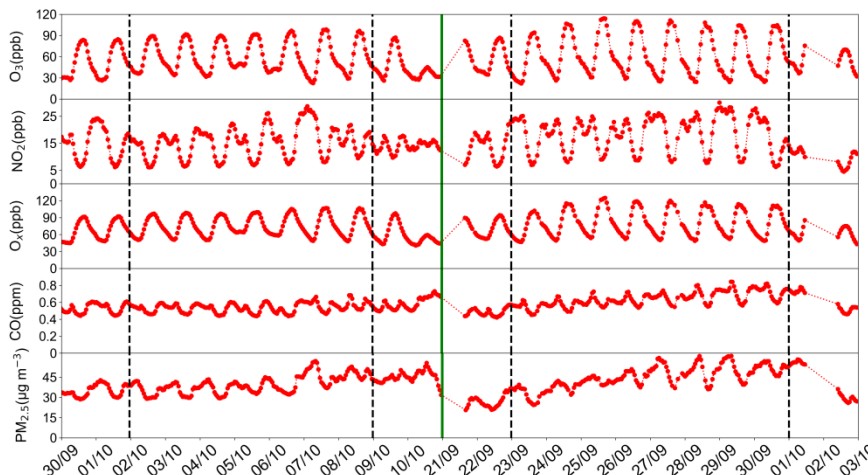

**Figure 3: Hourly surface concentrations of pollutants during the study period. The green line is added to separate the two episodes, the black dashed lines indicate the two days before and after the episodes.**



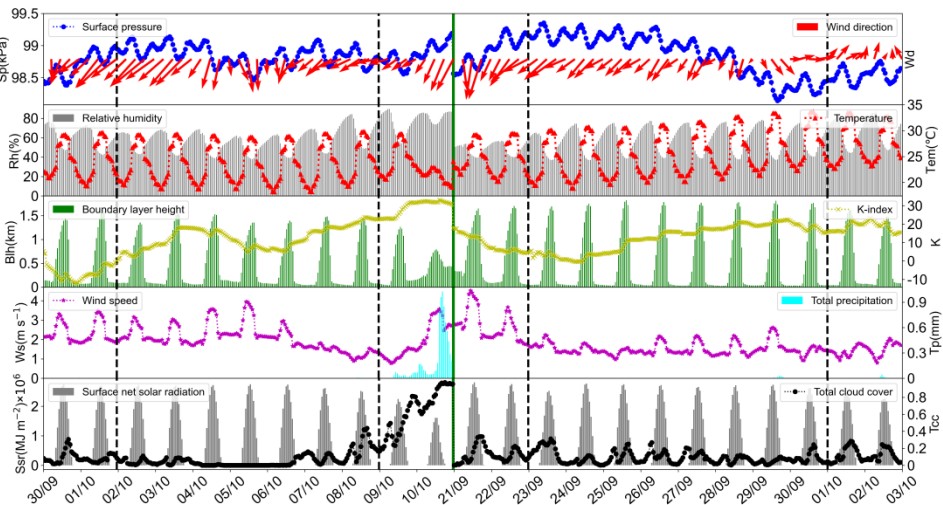


**Figure 4: Same as Fig. 3 except for meteorological parameters.**



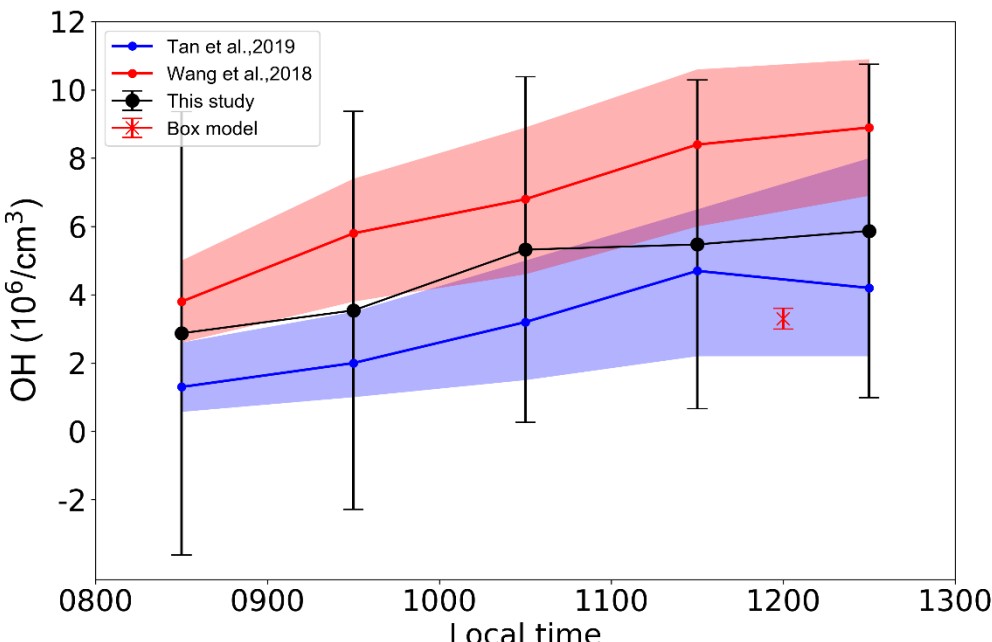

**Figure 5: Hourly average OH concentrations between 08:00 and 13:00 derived from the OBM are shown in black line with black dots, observed OH concentrations by Tan et al. (2019) are shown in blue line and blue shade, calculated OH concentrations by Wang et al. (2018) are shown in red line and red shade. The blue shade denotes the 25% and 75% percentiles of the data, the red shade indicates the 95% C. I. of the data. The red cross with red vertical whiskers denotes the mean OH concentration and 1-standard deviation, respectively, calculated by a box model constrained by observed ambient conditions observed during the two episodes.**


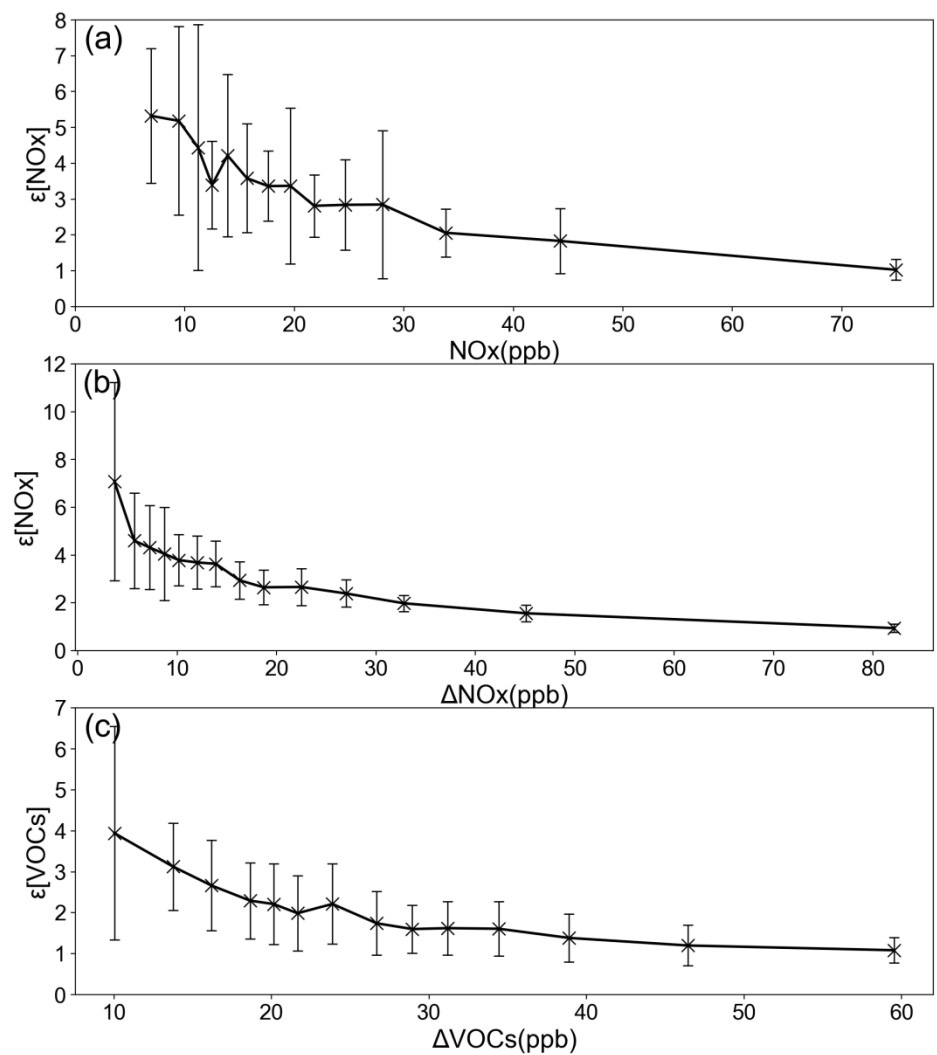


**Figure 6: Ozone production efficiency plotted as a function of NOₓ (a), oxidized NOₓ (b), and oxidized VOC (c).**





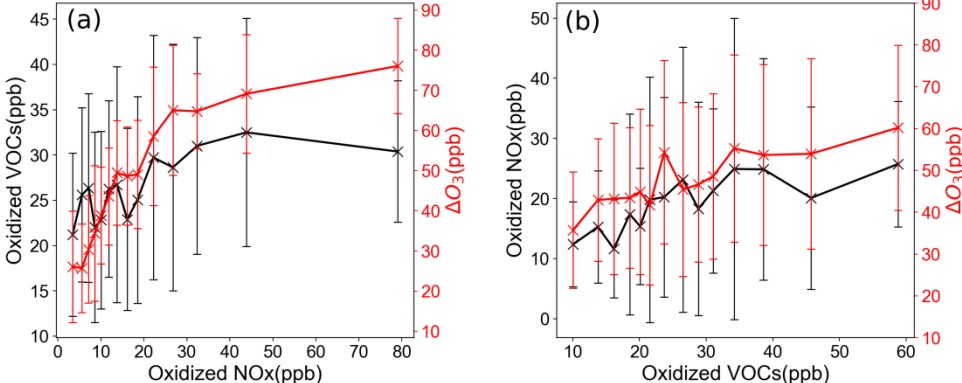

**Figure 7: Ozone formation rate (ΔO₃, right hand side in red) and the oxidized VOC (left hand side in black) plotted as a function of oxidized NOₓ (a). Ozone formation rate (ΔO₃, right hand side in red) and oxidized NOₓ (left hand side in black) plotted as a function of the oxidized VOC (b).**



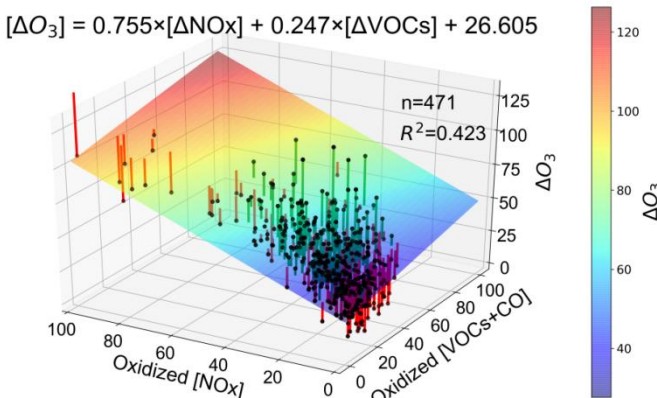

$$[\Delta O_3] = 0.755 \times [\Delta NOx] + 0.247 \times [\Delta VOCs] + 26.605$$

**Figure 8: Three-dimensional depiction of ozone formation rate ($\Delta O_3$, z-axis) plotted as a function of oxidized NO$_x$ (x-axis) and oxidized VOC (y-axis). The black dots denote values of $\Delta O_3$, the colored plane is the best linear fit to the black dots, and the green and red bars denote positive and negative deviations from the plane, respectively. The equation listed represents the surface as a function of oxidized NO$_x$ and oxidized VOC. $R^2$ is the square of correlation coefficient of the linear regression.**

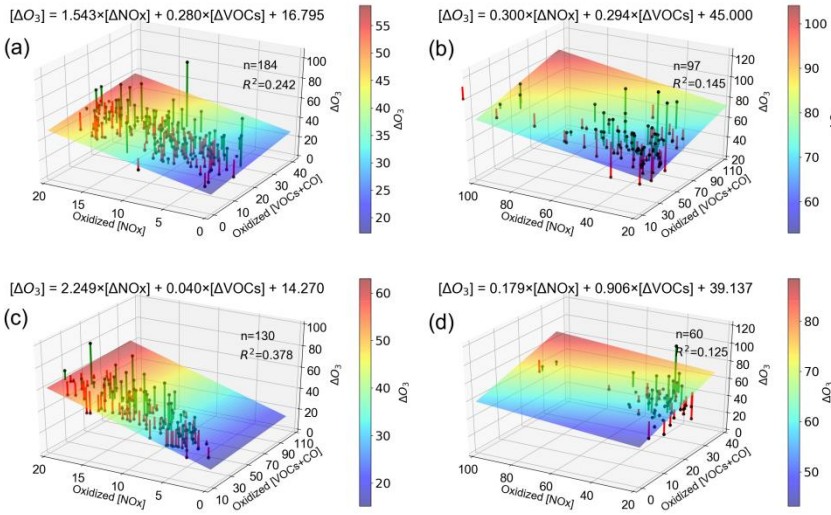

**Figure 9: Same as Fig. 8 except divided into four quadrants: (a) low NOx and low VOC (NOx<20 ppb, VOC<25 ppb), (b) high NOx and high VOC (NOx>20 ppb, VOC>25 ppb), (c) low NOx and high VOC (NOx<20 ppb, VOC>25 ppb), (d) high NOx and low VOC (NOx>20 ppb, VOC<25 ppb).**





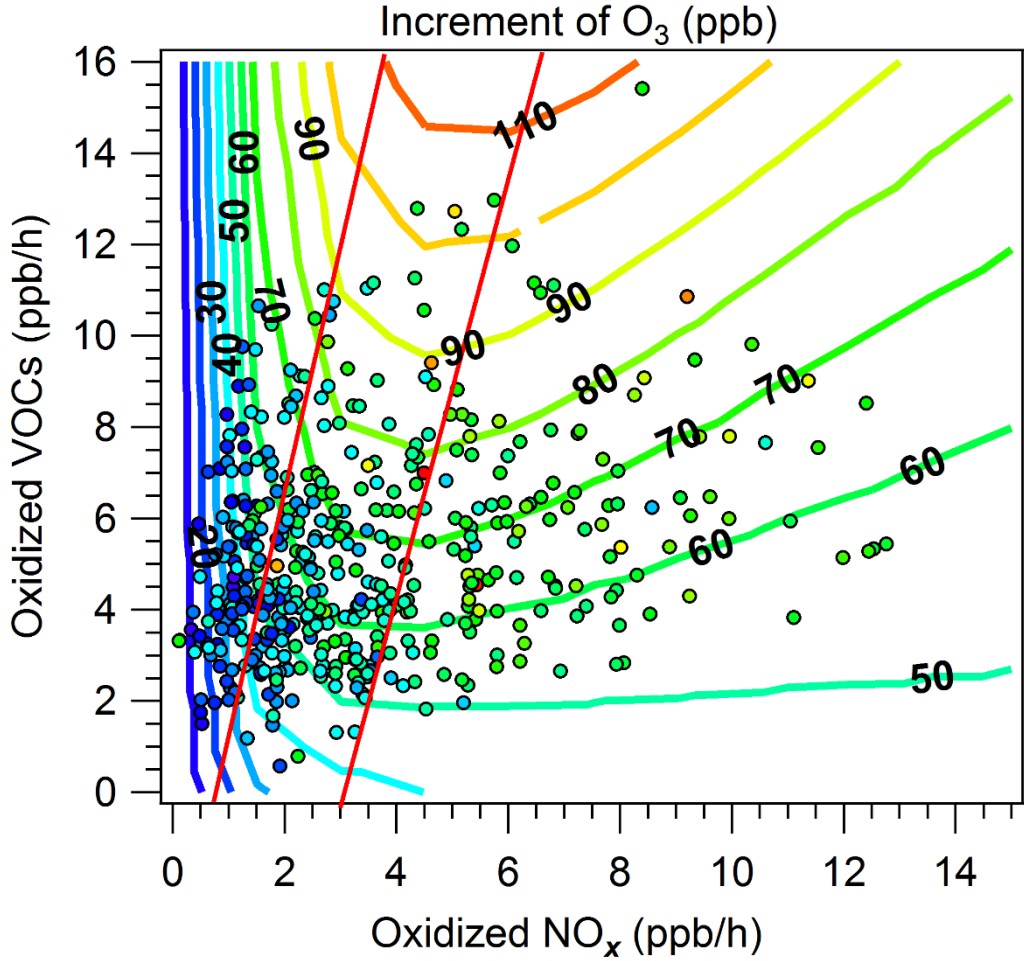

**Figure 10: Ozone isopleths (in ppbv) of traditional 2D-EKMA plot for the two episodes calculated by the box model are shown in colored lines, ozone concentrations at 13:00 local time derived by the OBM are shown in colored dots, x-axis and y-axis are hourly oxidized NO$_x$ and oxidized VOC, respectively.**
