# Peer review of "Observation-based Analysis of Ozone Production Sensitivity for Two Persistent Ozone Episodes in Guangdong, China"

_Atmospheric Chemistry and Physics, 2022_

## Author Comment (AC1)

Dear Editor,

We appreciate the prompt reviews and would like to thank the reviewers for insightful comments and suggestions on our manuscript entitled "Observation-based Analysis of Ozone Production Sensitivity for Two Persistent Ozone Episodes in Guangdong, China" (MS No.: acp-2022-50). We have carefully considered all comments and suggestions. Listed below are our point-by-point responses to all comments and suggestions of this reviewer (Reviewer's points in black, our responses in blue).

**Anonymous Referee #1**

The authors developed an observation-based method to investigate the ozone production efficiency and ozone production sensitivity to precursors for two persistent ozone episodes in Guangdong, China, based on the hourly surface $O_3$, $PM_{2.5}$, CO and $NO_2$ data at 77 stations in Guangdong during the period 2018-2019. They also performed a box model constrained by ambient conditions observed during the two episodes for comparison. They find 67% of the station-days exhibit ozone formation sensitivity to NOx, which differs from other previous studies which suggested that limiting VOC emission rather than NOx would be more effective in reducing ozone in Guangdong, and these results are in semi-quantitative agreement with the results calculated by box model. The authors make some arbitrary assumptions and simplifications in derivations of VOC and OH concentrations, which is a major weakness of the current work. I had a number of specific comments for the authors to consider and address before publication.

**Response:**

We acknowledge this referee's concerns that we have made "some arbitrary assumptions and simplifications in derivations of VOC and OH concentrations". We address these assumptions and simplifications individually in the following.

Specific comments:

1. Specific information on the ratios of VOC/CO that is used in this study for the derivation of VOC is better added in the SI. Also, the uncertainties due to this treatment of VOC on OH concentration should be discussed.

**Response:**

Thank you for the suggestion. We now have added in the SI a table of formula on how VOCs are derived from the ratios of VOC/CO, including the fresh CO, leftover CO and OVOCs. The uncertainties due to this treatment of VOC on OH concentration are now presented in Section 3.5 by replacing most of Line 311–315 with the following paragraph.

The second largest source of uncertainty is the evaluation of VOCs. Individual VOCs, including OVOCs, are calculated based on the observed concentration of CO and the ratio of VOC/CO in the emission inventories as discussed in Section 2.2.2. We have evaluated the VOCs and OVOCs derived this way by comparing their contributions to the OH reactivity observed by Tan et al. (2019) in PRD in autumn 2014. There is a reasonable agreement between our estimates of the contributions of NOx, CO, OVOCs and VOCs to the OH reactivity and those of Tan et al. (2019) except for a 35% underestimation of VOCs. Hence we estimate the uncertainty in the evaluation of VOCs to be in the range of 30-50%.

2. Line 80-81, the author uses the same way to evaluate the leftover VOC as to evaluate the leftover CO in the following day. However, VOCs can continue to be oxidized by OH and $NO_3$ in the afternoon and at night. How great are the effects of this neglect of the depletion of leftover VOC on the derived VOC and OH concentrations in the following day?

**Response:**

The referee makes a very insightful point on "VOCs can continue to be oxidized by OH and NO3 in the afternoon and at night". We now clarify in the SI how the leftover

VOCs are evaluated. By assuming the CO at 13:00 to be leftover CO, we imply that the leftover CO includes all CO emitted more than 24 hours ago. Hence the oxidation of VOCs in the afternoon of the previous day is included. Yes, we have neglected the oxidation by $NO_3$. This is because we believe $NO_3$ concentration is suppressed by the heterogeneous reaction of $N_2O_5$ with aerosols under the humid and polluted environment of the Pearl River Delta (PRD). Since the contribution of leftover VOCs to total VOCs is only 14%, neglecting the oxidation by $NO_3$ should have less than 2% effect on the total VOCs derived by our method. Hence we estimate that the impact of neglecting $NO_3$ on the OH calculated by the box model (Figure 4, red cross with 1-sigma bar) is less than 1%. The OH derived by the observation-based method (OBM) depends mainly on the observed ratio of $NO_2/CO$ between 08:00 and 13:00, the effect of VOC on OH derived from the OBM is implicitly included.

3. Line 81-82, the authors state that the oxidized VOC are estimated from the observed ratios of HCHO, $CH_3CHO$, and ketone to CO in Wang et al., 2016, and other OVOCs are not included. What is the basis for this treatment? Besides, HCHO, $CH_3CHO$, and ketone are not only photochemical products of VOC oxidation but also from direct anthropogenic emission. How do the authors deal with the difference in emission-related origin of OVOC among different locations? Given that OVOCs typically make large portion of OH reactivity, the estimations of OVOCs are crucial for the simulation of OH concentrations. How large are the uncertainties of these assumption on the predicted OH concentrations?

**Response:**

Some issues raised in this comment are addressed in our response to Comment #1. The referee makes an important comment that OVOCs typically make large portion of OH reactivity while our treatment of OVOCs basing on the observed ratio of OVOC to CO may have large uncertainty. Yes, in our study the portion is about 24% vs. 26% for the portion of VOCs (see line 88-95 of the revised manuscript, also copied below), mostly from HCHO and $CH_3CHO$. Since at least one aldehyde is produced during the

complete oxidation process of each VOC molecule to $CO/CO_2$ and $H_2O$, this 24% portion of OH reactivity derived by our method is consistent with this notion when the loss of VOCs to aerosols and/or other sinks is taken into consideration. We estimate the OVOCs by using their ratios to CO observed in PRD by Wang et al. (2016). This method has the advantage that it includes the direct anthropogenic emission of OVOCs as well as the photochemical production of OVOCs from VOC oxidation, although the observed ratio may not represent the ratios at individual stations. We acknowledge that our treatment may introduce significant uncertainty to the estimated overall VOCs, which can affect the OH calculated in the box model by as much as 20% (Figure 4, red cross with 1-sigma bar). Nevertheless, the uncertainty has no direct effect on the OH derived by OBM because the effect is implicitly included as mentioned above.

We further address this comment by adding line 88-95 to the revised manuscript which is shown in the following. "The VOCs and OVOCs derived this way can be validated by comparing with observed values in terms of the OH reactivity. Tan et al. (2019) reported that observed NOx, CO, HCHO and VOCs in PRD in autumn 2014 contributed, respectively, 14%, 10%, 5–8% and 20%, for a total of about 50% to the observed OH reactivity, which scale to 28%, 20%, 10–16% and 40%, respectively when normalized to 100%. In comparison, in our study the average NOx, CO, OVOCs and VOCs contribute 33%, 17%, 24% and 26%, respectively to the OH reactivity. There is a reasonable agreement between our results and those of Tan et al. (2019) except for OVOCs and VOCs. The disagreement on OVOCs can be easily explained by the fact that HCHO accounts for about two thirds of the OVOCs in our case. Nevertheless, the underestimate of the VOC contribution in our study remains unresolved and suggests significant uncertainty in the VOCs derived by our method."

4. Line 85-112, the derivation of OH concentrations and calculation of oxidized VOC and NOx in this study are all based on the Lagrangian condition assumption, which rarely exists in the real atmosphere, so the authors make a selection criterion to filter out days satisfy the quasi-Lagrangian condition. What is the basis of this selection

criterion?

**Response:**

The referee is right that the Lagrangian condition rarely exists. That is why we select the quasi-Lagrangian condition of 50% of 1-standard deviation (line 110) as the selection criterion. The basis of this selection criterion can be seen in the fourth panel of Figure 2 (CO profile), where the values of 50% of 1-standard deviation at 08:00 and 13:00 are reasonably close to the average values (black dots) which are presumed to represent the Lagrangian condition. This criterion usually filters out about 60% of data. We have tested this selection criterion by parameterizing it between 30% to 80% of 1-standard deviation and found our major results are robust within this range. Thanks to the referee's comment, we now have added the1-standard deviations to Figure 2 and inserted the previous sentence to line 112 of the revised manuscript.

5. Line 122-126, why the product of the average OH at noontime and the mean NOx in 13:00-16:00 can be used as the hourly NOx emission rate between 08:00 and 13:00?

**Response:**

We assume that the quasi-steady state of NOx in 13:00–16:00 in Figure 2 is maintained by the balance between the oxidation loss of NOx and its emission. This assumption is based on the notion that oxidation of $NO_2$ by OH is the predominant sink of NOx in 13:00–16:00, of which the integration over the mixed/boundary layer should be balanced by the emission flux of NOx according to the continuous equation of NOx. Assuming the oxidation loss rate of NOx within the mixed layer is uniform with height (supported by models), we obtain that the divergence of the hourly NOx emission rate is equal to the oxidation loss rate of NOx in 13:00–16:00. Finally, we assume the hourly NOx emission rate in 13:00–16:00 can be used for 08:00–13:00, i.e. neglecting the variation in NOx emission between 08:00 and 13:00. We have added the statements above to line 137 in the revised manuscript.

6. Line 311-315, these sentences are totally a copy of the sentences in line 78-83, and do not provide any useful information, and I would like to suggest the authors to delete these sentences, and provide more useful information about the uncertainty analysis.

**Response:**

We accept the referee's criticism and suggestion. In the revised manuscript the cited sentences have been replaced with the followings.

Individual VOCs, including OVOCs, are calculated based on the observed concentration of CO and the ratio of VOC/CO in the emission inventories as discussed in Section 2.2.2. We have evaluated the VOCs and OVOCs derived this way by comparing their contributions to the OH reactivity observed by Tan et al. (2019) in PRD in autumn 2014. There is a reasonable agreement between our estimates of the contributions of NOx, CO, OVOCs and VOCs to the OH reactivity and those of Tan et al. (2019) except for a 35% underestimation of VOCs. Hence we estimate the uncertainty in the evaluation of VOCs to be in the range of 30–50%.

7. I would like to suggest the authors can review other literatures reporting the ozone production efficiency in PRD areas to strengthen the discussion.

**Response:**

Thank you for this helpful suggestion. It is an embarrassing oversight that we didn't review previous works on the ozone production efficiency in PRD areas. We have added the following discussion at line 192.

Compare to previous investigations in PRD areas, values in Fig. 5a at NOx higher than 20 ppb are in good agreement with the ε(NOx) values of 2.1–2.5 found at urban stations in PRD by Yu et al. (2020) and Lu et al. (2010b). However, ε(NOx) values of 6.0–13.3 were found at rural stations in PRD (Lu et al., 2010b; Wei et al., 2012; Xu et al., 2015; Yang et al., 2017), which are about a factor of 2 higher than our values at

low NOx. Considering that our values are derived for two ozone pollution episodes in which the ε(NOx) should be higher than non-episode periods, this discrepancy is puzzling.

8. Considering that all the VOCs data are estimated based on the ratios of VOC/CO in the emission inventory, I think it is not appropriate to name it an observation-based method.

**Response:**

The derivation of OH by OBM, which is a critical part of this study, depends entirely on observed ratio of $NO_2/CO$ between 08:00 and 13:00, no VOC is needed. So at least in this part of the study, we think it is appropriate to name it an observation-based method. We did acknowledge in the paper that VOCs played important roles in the part of ozone formation and there were significant uncertainties in our estimate of VOCs. Nevertheless, the ozone formation analysis part of our study was based extensively on the observed data of NOx, $O_3$ and CO. Therefore, with due respect, we believe it is appropriate to name our study an observation-based method.

**References**

Lu, K., Zhang, Y., Su, H., Shao, M., Zeng, L., Zhong, L., Xiang, Y., Chang, C. C., Chou, C. K. C., and Andreas, W.: Regional ozone pollution and key controlling factors of photochemical ozone production in Pearl River Delta during summer time, Sci. China Chem., 53, 651–663, https://doi.org/10.1007/s11426-010-0055-6, 2010.

Tan, Z., Lu, K., Hofzumahaus, A., Fuchs, H., Bohn, B., Holland, F., Liu, Y., Rohrer, F., Shao, M., Sun, K., Wu, Y., Zeng, L., Zhang, Y., Zou, Q., Kiendler-Scharr, A., Wahner, A., and Zhang, Y.: Experimental budgets of OH, $HO_2$, and $RO_2$ radicals and implications for ozone formation in the Pearl River Delta in China 2014, Atmos. Chem. Phys., 19, 7129–7150, https://doi.org/10.5194/acp-19-7129-2019, 2019.

Wang, B., Liu, Y., Shao, M., Lu, S., Wang, M., Yuan, B., Gong, Z., He, L., Zeng, L.,

Hu, M., and Zhang, Y.: The contributions of biomass burning to primary and secondary organics: A case study in Pearl River Delta (PRD), China, Sci. Total Environ., 569–570, 548–556, https://doi.org/10.1016/j.scitotenv.2016.06.153, 2016.

Wei, X., Liu, Q., Lam, K. S., and Wang, T.: Impact of precursor levels and global warming on peak ozone concentration in the Pearl River Delta region of China, Adv. Atmos. Sci., 29, 635–645, https://doi.org/10.1007/s00376-011-1167-4, 2012.

Xu, Z., Xue, L., Wang, T., Xia, T., Gao, Y., Louie, P. K. K., and Luk, C. W. Y.: Measurements of peroxyacetyl nitrate at a background site in the Pearl River Delta region: Production efficiency and regional transport, Aerosol Air Qual. Res., 15, 833–841, https://doi.org/10.4209/aaqr.2014.11.0275, 2015.

Yang, Y., Shao, M., Keβel, S., Li, Y., Lu, K., Lu, S., Williams, J., Zhang, Y., Zeng, L., Nölscher, A. C., Wu, Y., Wang, X., and Zheng, J.: How the OH reactivity affects the ozone production efficiency: case studies in Beijing and Heshan, China, Atmos. Chem. Phys., 17, 7127–7142, https://doi.org/10.5194/acp-17-7127-2017, 2017.

Yu, D., Tan, Z., Lu, K., Ma, X., Li, X., Chen, S., Zhu, B., Lin, L., Li, Y., Qiu, P., Yang, X., Liu, Y., Wang, H., He, L., Huang, X., and Zhang, Y.: An explicit study of local ozone budget and NOx-VOCs sensitivity in Shenzhen China, Atmos. Environ., 224, 117304, https://doi.org/j.atmosenv.2020.117304, 2020.

---

## Author Comment (AC2)

Dear Editor,

We appreciate the prompt reviews and would like to thank the reviewer for insightful comments and suggestions on our manuscript entitled "Observation-based Analysis of Ozone Production Sensitivity for Two Persistent Ozone Episodes in Guangdong, China". We have carefully considered all comments and suggestions. Listed below are our point-by-point responses to all comments and suggestions of this reviewer (Reviewer's points in black, our responses in blue).

Comments to the Author

In this paper by Song et al., the authors use 2 episodes in Guangdong, China and a large number of measurement sites in the vicinity to construct an observation-based method (OBM), utilizing the measurements of various pollutants from said sites alongside a box model based on the CB05 chemical mechanism, with the purpose of determining ozone production efficiency (OPE) from NOx and VOCs. They conclude that the area is under a NOx limited regime, indicating that limiting NOx emissions is the optimal strategy to reduce ozone formation in the area, contrary to previous studies.

While the paper does have its strong points – the analysis is thorough, the English used is clear and appropriate – it is not without shortcomings, many of which reviewer #1 covered. The paper merits publication based on the rigor of its analysis, but not that of the conclusions. As such I recommend the paper for publication only after the following points have been addressed and the discussion strengthened.

**Response:**

We appreciate the insightful comments and suggestions.

**Science comments:**

1. As the authors mention two episodes are not enough. In addition, they are well into

the ozone season in the fall, which could further bias the results. For example, biogenic emissions of VOCs are going to be significantly less than what they would be during the summertime, which could tip the balance of the OPE. A section should be added to discuss the potential differences between summer and fall months. The box model the authors have developed can be used, driven with meteorological variables from the observation sites during different seasons (if available), to investigate

**Response:**

Thank you for the very helpful comments and suggestions here and in Editorial comment 3. In response a new Fig. 1b is added which shows the average ozone concentrations of all ozone exceeding days in Guangdong in 2018 and 2019. One can see clearly that the ozone distribution during the two episodes in autumn is representative of and even slightly higher than the ozone concentrations during ozone pollution days in Guangdong in the entire two years. In fact, the monthly peak ozone concentration in Guangdong tend to occur in September and October because Guangdong is usually under heavily overcast conditions with prevailing southerly winds bringing clean moist air from the South China Sea in the summer which tends to suppress the ozone formation. We have added some discussions on this point around line 58 of the revised manuscript.

2. Based on (1), the usage of CO to VOC ratios, while a valid strategy for anthropogenic emissions completely neglects possible biogenic impacts and thus is better suited towards the urban sites much more than the rural sites. In addition to the current analysis, it would be of value that the authors also conduct the same by splitting the sites in rural and urban which would be more representative

**Response:**

We are conducting a follow up study by splitting the sites in rural and urban. In regard to possible biogenic impacts, we have difficulty accounting for the effect of isoprene

because it is highly variable due to its short lifetime. Nevertheless, part of the biogenic impact (up to one half) is included in our estimate of HCHO, $CH_3CHO$ and ketones. For instance, we use observed ratio of HCHO/CO to evaluate HCHO which includes HCHO produced from biogenic VOCs. Furthermore, we have added some discussions on the uncertainty of VOCs and OVOCs around line 88 of the revised manuscript, which are copied below.

The VOCs and OVOCs derived this way can be validated by comparing with observed values in terms of the OH reactivity. Tan et al. (2019) reported that observed NOx, CO, HCHO and VOCs in PRD in autumn 2014 contributed, respectively, 14%, 10%, 5–8% and 20%, for a total of about 50% to the observed OH reactivity, which scale to 28%, 20%, 10–16% and 40%, respectively when normalized to 100%. In comparison, in our study the average NOx, CO, OVOCs and VOCs contribute 33%, 17%, 24% and 26%, respectively to the OH reactivity. There is a reasonable agreement between our results and those of Tan et al. (2019) except for OVOCs and VOCs. The disagreement on OVOCs can be easily explained by the fact that HCHO accounts for about two thirds of the OVOCs in our case. Nevertheless, the underestimate of the VOC contribution in our study remains unresolved and suggests significant uncertainty in the VOCs derived by our method.

3. I second reviewer's #1 comment about the NOx quasi steady state. This would only apply from 13:00 to 16:00. Using the average OH value for the early day is not accurate.

**Response:**

The following is our response to the corresponding comment of Referee #1:

We assume that the quasi-steady state of NOx in 13:00–16:00 in Fig.2 is maintained by the balance between the oxidation loss of NOx and its emission. This assumption is based on the notion that oxidation of $NO_2$ by OH is the predominant sink of NOx in 13:00–16:00, of which the integration over the mixed/boundary layer should be

balanced by the emission flux of NOx according to the continuous equation of NOx. Assuming the oxidation loss rate of NOx within the mixed layer is uniform with height (supported by models), we obtain that the divergence of the hourly NOx emission rate is equal to the oxidation loss rate of NOx in 13:00–16:00. Finally, we assume the hourly NOx emission rate in 13:00–16:00 can be used for 08:00–13:00, i.e. neglecting the variation in NOx emission between 08:00 and 13:00. We have added the statements above to line 137 in the revised manuscript.

4. The calculation of OPE assumes that the only real sink of NOx is the ozone formative chemistry. However, NOx is also lost to other processes and in an area like deposition and nitrate formation. The deposition is briefly mentioned towards the end, but some additional discussion and/or an estimate of the magnitude of the effect should be provided. Given the close proximity of ports in the area and therefore the likely high emissions of SO2 and subsequent sulfate formation, the additional NOx sinks could be of an important magnitude. I do realize that such an analysis would be out of the scope of the paper, and I do not require that authors conduct it, but some additional discussion on the matter is warranted, given the number of assumptions already used. On that note, particularities of Guangdong should be added to the introduction e.g., nearby ports, major highways, nearby agricultural activities etc.

**Response:**

We appreciate this important comment. Yes, we have neglected the heterogeneous reactions in this study. Since the effect of heterogeneous reactions is included in the observations, the neglection of heterogeneous reactions on NOx can lead to an overestimate of OH concentrations by the OBM which is a key product of this study. We have added in Section 3.5 of the revised manuscript the following paragraph on the effect of neglecting heterogeneous reactions.

Another source of uncertainty may come from the neglection of heterogeneous reactions in this study. The largest impact of neglecting heterogeneous reactions is

most likely to involve NOx between 08:00 and 13:00, during which the OH is derived. Since the effect of heterogeneous reactions is included in the observations, the neglection of any heterogeneous removal of NOx (e.g. deposition of NOx on aerosols in the humid conditions in Guangdong) can lead to an overestimate of OH concentrations by the OBM. This would have a significant impact on the outcome of this study, as OH plays a critical role in the photochemistry of NOx, VOCs and ozone. On the other hand, presence of significant natural sources of NOx such as biogenic emission and/or lightning source in 08:00–13:00 would lead to an underestimate of OH concentration.

**Editorial comments:**

1. The timeseries of meteorological parameters is more suited for the SI. Use the diurnal profiles instead in the manuscript, so the reader can directly go back and forth with the diurnal concentrations to clearly see the dilution effect due to the PBL.

**Response:**

Thanks. We have moved the timeseries of meteorological parameters to the SI in the revised manuscript.

2. While I do understand why Figure 8 was added, and it holds a lot of valuable information, it would be best to either omit it or add it to the SI. Figure 9 is more appropriate, and it would be even better if you turn it into 2D plots with variable marker sites

**Response:**

Thanks. We have moved Figure 8 to the SI in the revised manuscript.

3. The combined site isopleth could use some polishing; fill out the contours. Also, I very strongly recommend that you make isopleth for each of the observation site clusters from Figure 1. This also feeds into point 2 from the science comments above.

**Response:**

Thanks. We have improved Fig. 1 and added a new Fig. 1b to address this suggestion and those raised in point 1 of the science comments. For details please see our response to point 1 of the science comments.

---

## Author Response (AR2)

Dear Editor,

Thanks for your great efforts on our manuscript entitled "Observation-based Analysis of Ozone Production Sensitivity for Two Persistent Ozone Episodes in Guangdong, China" (MS No.: acp-2022-50).

As you suggested, we have deposited our data in figshare repository (https://doi.org/10.6084/m9.figshare.20055221), and revised the data availability section accordingly.

Regarding Figure 8, we have checked it using the Coblis-Color Blindness Simulator, and find that the color schemes used allow readers with color vision deficiencies to correctly interpret our findings.